*Perspective*

# Expansion omics: from expansion microscopy to spatial omics

Zhen Dong [1,2,3,4✉], Weirong Xiang[1,2,3,4], Wenhao Jiang [1,2,3] & Tiannan Guo [1,2,3✉]

## Abstract

Tissue expansion, originally developed for super-resolution imaging, has become a foundation for expansion omics (ExO), a growing field that uses physical tissue expansion to enable spatially resolved omics profiling. In this perspective, we explore how ExO integrates multi-omics through chemical anchoring strategies that ensure selective retention of diverse molecular species, together with improved spatial resolution from the subcellular resolution for profiling to the sub-nanometer scale for imaging, allowing precise detection of biomolecules and their link with biological function. These capabilities have empowered tissue expansion to be successfully applied across multiple spatial omics modalities, including epigenomics, transcriptomics, proteomics, and lipidomics, enabling high-resolution mapping of chromatin states, gene expression, protein localization, and lipid distributions. Moreover, ExO supports spatial multi-omics approaches that jointly capture and correlate multiple biomolecular dimensions within the same tissue context. However, challenges remain in expansion resolution, molecular retention, hydrogel adaptability, data scalability, and AI-driven analysis. As tissue expansion evolves, its integration of super-resolution imaging and spatial omics establishes it as a core technology for whole-slide, single-cell multi-omics and the development of the Artificial Intelligence Virtual Cell, advancing spatial biology and medicine.

## Introduction

Expansion microscopy (ExM) is a transformative technique that physically enlarges biological specimens via tissue expansion using swellable hydrogels to overcome the diffraction limit and enable nanoscale imaging with conventional antibodies and microscopes (Chen et al, 2015; Chozinski et al, 2016; Tillberg et al, 2016; Wassie et al, 2019). Progress in ExM methodology, including higher expansion factors to increase detection resolution dramatically (Chang et al, 2017; Louvel et al, 2023; Sarkar et al, 2022; Wang et al, 2024) and optimized labeling strategies for selective molecular retention (Klimas et al, 2023; Sun et al, 2021), has established ExM as a powerful tool for linking nanoscale organization to biological function. For example, ExM has been used to map chromatin architecture at the nanoscale (Pownall et al, 2023), resolve neuronal and synaptic structures in the brain (Gao et al, 2019; Klimas et al, 2023), and provide insights into protein conformations (Shaib et al, 2024), revealing structural-functional relationships. These studies highlight ExM's critical role in uncovering the functional implications of nanoscale organization in biology.

The process of tissue expansion involves several key steps (Fig. 1a), beginning with biomolecule anchoring. Proteins are typically anchored by introducing acryloyl groups to lysine residues via N-hydroxysuccinimide (NHS) ester-based anchors (Chozinski et al, 2016; Li et al, 2022; Tillberg et al, 2016) or through formaldehyde-mediated acrylamide-assisted protein retention (Ku et al, 2016). RNA anchoring can be achieved via direct chemical modification (e.g., LabelX, Mel-phaX) or polymer-linkable FISH probes such as acrydite/poly(T) oligonucleotides and TRITON-modified sequences (Wen et al, 2023). Lipids are retained either covalently through gel-linkable acrylates to lipid probes (Götz et al, 2020; Shin et al, 2025; Wen et al, 2020; White et al, 2022) or non-covalently via hydrophobic/electrostatic interactions with anchored membrane proteins (Chan et al, 2024; Hung et al, 2024). Universal anchoring strategies include metabolic labeling-based click chemistry (Sun et al, 2021) and methacrolein-based direct grafting (Klimas et al, 2023). Next, gelation occurs through free-radical polymerization, forming a polyacrylate hydrogel. This process is typically initiated by a radical initiator, such as ammonium persulfate, and accelerated by a catalyst, like $N,N,N',N'$-tetramethylethylenediamine (TEMED). The polymerization results in a robust three-dimensional hydrogel network, commonly composed of monomers such as sodium acrylate, co-monomers like acrylamide, and crosslinkers such as bisacrylamide. These components facilitate biomolecule anchoring and ensure uniform hydrogel swelling, preserving spatial structure and enhancing resolution. After gelation, homogenization is needed to disrupt structures that might otherwise impede isotropic expansion. Depending on the sample type, this step involves enzymatic digestion (e.g., Proteinase K/LysC) (Tillberg et al, 2016), heat-induced SDS denaturation (Gambarotto et al, 2019; Klimas et al, 2023), or harsher treatments such as autoclaving (Valdes et al, 2024) and microwave irradiation (Guo et al, 2025a). Tougher tissues often require stronger homogenization to achieve uniform expansion. Following expansion, detection can be performed through fluorescence visualization (the most widely used) or chromogen deposition (M'Saad et al, 2022), while molecular profiling can be achieved through

[1]Affiliated Hangzhou First People's Hospital, State Key Laboratory of Medical Proteomics, School of Medicine, Westlake University, Hangzhou, Zhejiang Province, China. [2]Westlake Center for Intelligent Proteomics, Westlake Laboratory of Life Sciences and Biomedicine, Hangzhou, Zhejiang Province, China. [3]Research Center for Industries of the Future, School of Life Sciences, Westlake University, Hangzhou, Zhejiang Province, China. [4]These authors contributed equally: Zhen Dong, Weirong Xiang. ✉E-mail: dongzhen@westlake.edu.cn; guotiannan@westlake.edu.cn

https://doi.org/10.1038/s44320-025-00171-9 | Published online: 1 December 2025

**Glossary**

| | |
|---|---|
| **AcX** | Acryloyl-X, SE |
| **AI** | Artificial Intelligence |
| **AIVC** | Artificial Intelligence Virtual Cell |
| **ChromExM** | Chromatin Expansion Microscopy |
| **Click-ExM** | Click-Expansion Microscopy |
| **DestVI** | Deconvolution of Spatial Transcriptomics Profiles using Variational Inference |
| **Dual-ExM** | Dual-Expansion Microscopy |
| **ExEpi** | Expansion Microscopy for Epigenetics |
| **ExFISH** | Expansion Fluorescence In Situ Hybridization |
| **ExIMS** | Expansion Imaging Mass Spectrometry |
| **ExM** | Expansion Microscopy |
| **Ex-MSI** | Expansion Mass Spectrometry Imaging |
| **ExO** | Expansion Omics |
| **ExPRESSO** | Expand and comPRESS hydrOgels |
| **ExR** | Expansion Revealing |
| **ExSeq** | Expansion Sequencing |
| **Ex-ST** | Expansion Spatial Transcriptomics |
| **FAXP** | Filter-Aided Expansion Proteomics |
| **FLARE** | Fluorescent Labeling of Abundant Reactive Entities |
| **GA** | Glutaraldehyde |
| **GAMSI** | Gel-Assisted Mass Spectrometry Imaging |
| **GMA** | Glycidyl Methacrylate |
| **HCR** | Hybridization Chain Reaction |
| **iExM** | Iterative Expansion Microscopy |
| **IMC** | Imaging Mass Cytometry |
| **LCM** | Laser Capture Microdissection |
| **LC-MS/MS** | Liquid Chromatography-Tandem Mass Spectrometry |
| **LExM** | Lipid Expansion Microscopy |
| **LIGER** | Linked Inference of Genomic Experimental Relationships |
| **lncRNAs** | Long Noncoding RNAs |
| **moscot** | Multi-omics Single-cell Optimal Transport |
| **MERFISH** | Multiplexed Error-Robust Fluorescence In Situ Hybridization |
| **MIBI** | Multiplexed Ion Beam Imaging |
| **MS** | Mass Spectrometry |
| **MSI** | Mass-Spectrometry Imaging |
| **multiExR** | Multiplexed Expansion Revealing |
| **NHS** | N-hydroxysuccinimide |
| **NovoSpaRc** | de novo Spatial Reconstruction |
| **NSA** | N-succinimidyl Acrylate |
| **ONE** | One-Step Nanoscale Expansion |
| **PLATO** | Parallel-Flow Projection and Transfer Learning across Omics |
| **proExM** | Protein Expansion Microscopy |
| **ProteomEx** | Expansion Proteomics |
| **RCA** | Rolling Circle Amplification |
| **S4P** | Sparse Sampling Strategy for Spatial Proteomics |
| **SANTO** | coarSe-to-fine AligNment and sTitching for spatial Omics |
| **SCEPTRE** | Single Cell Evaluation of Post-TRanslational Epigenetic Encoding |
| **Seq-Scope-X** | Seq-Scope-Expanded |
| **SIMO** | Spatial Integration of Multi-Omics |
| **STalign** | Spatial Transcriptomics Align |
| **STIM** | Spatial Transcriptomics Imaging Framework |
| **TEMI** | Tissue-expansion Mass-spectrometry Imaging |
| **TREx** | Ten-Fold Robust Expansion Microscopy |
| **umExM** | Ultrastructural Membrane Expansion Microscopy |
| **uniExM** | United ExM |

next-generation sequencing (Fan et al, 2023), mass-spectrometry imaging (MSI) (Bai et al, 2023; Chan et al, 2024), or liquid chromatography-tandem mass spectrometry (LC-MS/MS) (Dong et al, 2024; Li et al, 2022).

Beyond imaging, ExM's ability to retain nucleic acids, proteins, lipids, and other biomolecules has opened new possibilities for spatial omics. This molecular retention has elevated ExM from a structural imaging technique to a versatile platform for single- and multi-omics analyses on expanded tissues, while preserving relative spatial organization. While recent reviews have thoroughly explored its biological applications (Hümpfer et al, 2024), chemical strategies (Wen et al, 2023), and biomedical potential (Jia et al, 2024), its full potential as a spatial omics platform, particularly in the context of high-resolution molecular profiling, has yet to be comprehensively discussed. A forward-looking perspective on expansion omics (ExO) is therefore necessary to illuminate this emerging frontier, where spatial context converges with molecular profiling at unprecedented resolution. In this perspective, we explore how tissue expansion advances spatial omics, highlight emerging opportunities, address current challenges, and outline future directions.

# Unfolding the expansion omics landscape

## Expansion epigenomics

Understanding the spatial organization of epigenetic modifications is essential for deciphering how the epigenome influences development and disease within complex tissues (Lu et al, 2022; Schueder and Bewersdorf, 2022). ExM offers a powerful approach to visualize chromatin and epigenetic landscapes at nanoscale resolution by physically enlarging biological specimens through swellable hydrogels, overcoming the resolution and spatial-context limitations of conventional chromatin assays.

Building on ExM, three recent imaging-based methods have extended the capabilities of epigenetic profiling (Fig. 1b; Table 1). Single Cell Evaluation of Post-TRanslational Epigenetic Encoding (SCEPTRE) integrates ExM with immunofluorescence and DNA FISH to detect multiple histone modifications at specific, non-repetitive genomic loci. With ~75 nm spatial resolution after expansion, it currently supports up to 4-plex detection and could be further scaled up, with potential for higher-plex detection through sequential hybridization or integration with multiplexed

error-robust fluorescence in situ hybridization (MERFISH)-like strategies (Woodworth et al, 2021). Expansion microscopy for epigenetics (ExEpi) employs dual immunolabeling of chromatin readers and histone marks to analyze spatial co-localization in expanded nuclei. A 4× expansion improves confocal resolution to ~70–80 nm while preserving topographical features, enabling the inference of protein-modification affinities in situ (Acke et al, 2022). While both SCEPTRE and ExEpi have been primarily demonstrated in cultured cells, Chromatin Expansion Microscopy (ChromExM) extends expansion epigenetics to more complex in vivo systems. It integrates metabolic labeling of DNA/RNA with immunostaining and achieves ~15× linear expansion, corresponding to ~15 nm resolution on a confocal microscope and ~3 nm with stimulated emission depletion super-resolution microscopy, enabling state-dependent mapping of transcription factor-chromatin interactions at single-nucleosome resolution. It supports ~4-plex imaging and has been successfully applied to developing zebrafish embryos (Pownall et al, 2023).

Together, these methods represent a technological progression from static, locus-specific histone profiling to nanoscale mapping of chromatin-associated proteins

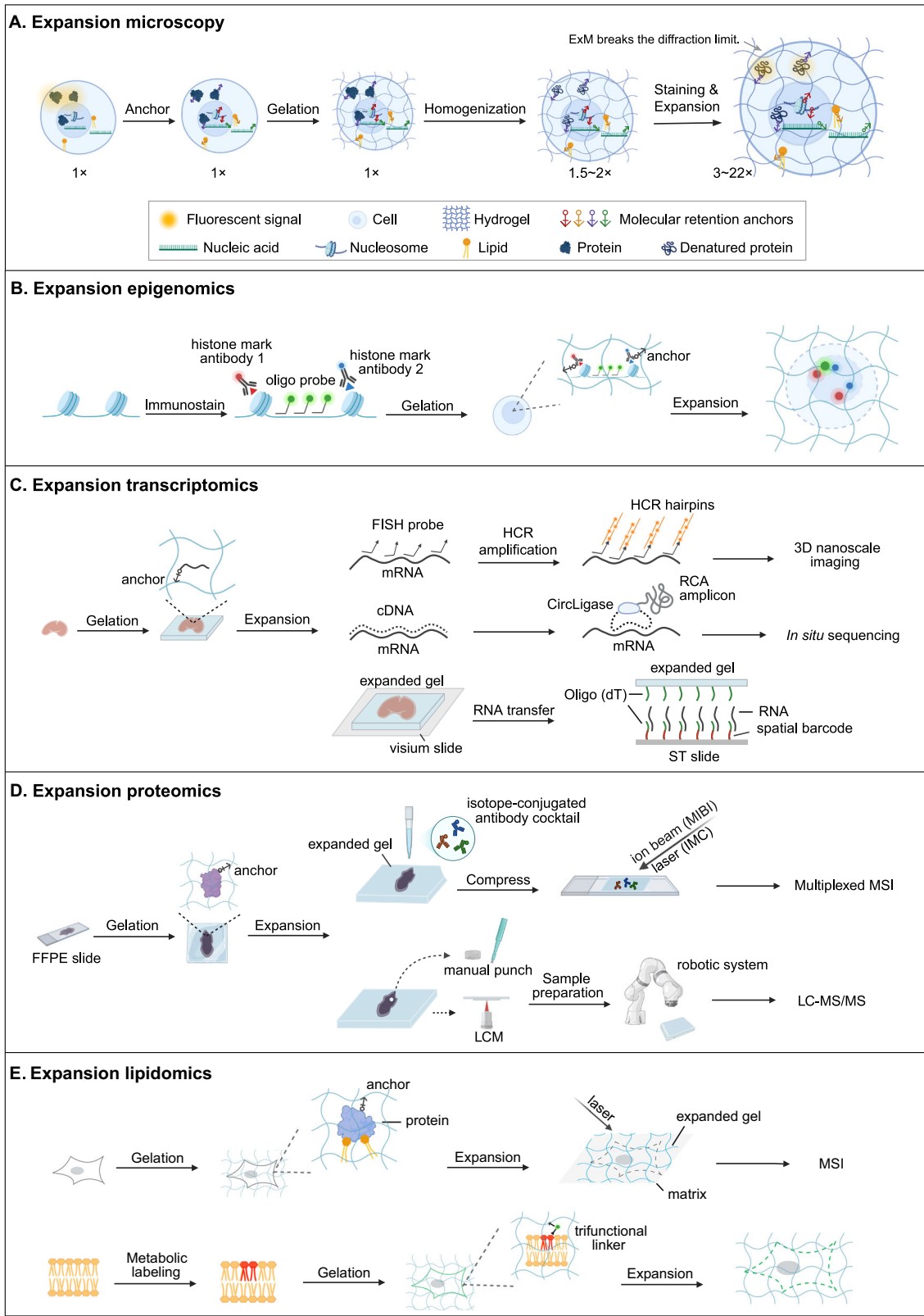

**Figure 1. Expansion microscopy workflow and applications in expansion omics.**

This figure illustrates the key steps of expansion microscopy (**A**), in which tissues are physically enlarged via hydrogel embedding and isotropic expansion to enable nanoscale imaging with conventional microscopy. The workflow involves molecular anchoring, gelation to form a swellable hydrogel, mechanical or enzymatic homogenization to facilitate uniform expansion, and post-expansion staining. During this process, proteins (including histones in nucleosomes) become denatured but are retained through molecular anchors, nucleic acids remain preserved in linear form when anchored, and lipids are generally lost unless specifically anchored or stabilized through interactions with proteins. This enables high-resolution visualization and mapping of biomolecules beyond the diffraction limit. Representative applications of expansion omics (ExO) include expansion epigenomics (**B**), transcriptomics (**C**), proteomics (**D**), and lipidomics (**E**), where ExO enhances spatial resolution while preserving molecular context for detailed mapping across multiple biomolecular layers. Created with BioRender.com.

and transcriptional regulators across distinct cellular states. Several limitations remain to be addressed: current multiplexing capacity is limited to ~2–4 simultaneous targets per experiment; imaging throughput typically reaches only hundreds of cells, restricting population-scale analysis; and broader applicability to diverse tissue types still requires further validation and optimization.

## Expansion transcriptomics

Spatial transcriptomics enables spatially resolved visualization and quantitative analysis of gene expression in complex tissues (Jain and Eadon, 2024). Techniques include imaging-based multiplexed in situ hybridization, alongside sequencing-based approaches such as in situ sequencing and capture. ExM has extended each of these approaches to achieve higher spatial resolution through physical tissue decrowding (Fig. 1c; Table 1).

Expansion fluorescence in situ hybridization (ExFISH) covalently anchors RNA to the hydrogel and enables post-expansion FISH for single-molecule detection, hybridization chain reaction (HCR) amplification, and multiplexed imaging (Chen et al, 2016). By physically separating RNA molecules, ExFISH achieves super-resolution visualization of RNA structure and localization with standard diffraction-limited microscopes in thick specimens, and has been applied to localizing neural mRNAs, mapping RNA structures, and visualizing long noncoding RNAs (lncRNAs). In fact, most of the imaging-based transcriptomic methods have benefited from tissue expansion. A notable example is the integration of MERFISH with ExM. By physically decrowding transcripts, expansion MER-FISH achieves near 100% detection efficiency for a ~130-RNA library with molecular densities more than tenfold higher than previously reported, while preserving spatial fidelity (Wang et al, 2018). Expansion sequencing (ExSeq)

combines in situ sequencing with expansion to enable both transcriptome-wide and targeted profiling (Alon et al, 2021). RNA molecules are anchored within the hydrogel and amplified into spatially indexed amplicons, which are read out by iterative fluorescence sequencing. Depending on the mode, ExSeq can capture splice variants, transcription factors, and lncRNAs at nanoscale resolution, mapping transcripts across neuronal dendrites, spines, and tumor tissues. Expansion spatial transcriptomics (Ex-ST) improves resolution in array-based spatial transcriptomics. By decrowding tissues before transcript capture, Ex-ST increases effective resolution from 55 to 20 µm for higher-resolution mapping in tissues such as the mouse brain (Fan et al, 2023).

Though all methods are based on ExM for higher spatial resolution, they differ in implementation and application. ExFISH enables mapping of specific RNA localization through hybridization imaging, while expansion MERFISH enables efficient measurement of high-abundance, multiplexed RNA libraries by physically decrowding transcripts. Transcriptome-wide sequencing at nanoscale precision is achieved with ExSeq, whereas Ex-ST improves resolution for array-based spatial transcriptome capture. Nonetheless, several challenges remain: designing highly specific probes for high-plex detection is further complicated by the physical and chemical constraints of the expanded environment. Balancing expansion conditions to preserve both nucleic acid accessibility and isotropy requires careful optimization. RNA integrity may be compromised throughout the expansion process, potentially reducing hybridization efficiency. Additionally, signal dilution and spatial distortions may compromise quantification accuracy and spatial fidelity.

## Expansion proteomics

Spatial proteomics aims to map protein distributions across tissue architectures to

reveal cellular organization and interactions (Guo et al, 2025b). Conventional approaches include affinity-based imaging and MS-based proteomics, with the latter offering unbiased, large-scale protein quantification. Recent advances such as deep visual proteomics (DVP) integrate artificial intelligence (AI)-guided image analysis, laser capture microdissection (LCM), and high-sensitivity MS to profile specific cell types at single-cell resolution (Mund et al, 2022; Rosenberger et al, 2023; Zheng et al, 2025). However, LCM-based methods face limitations: regions smaller than 20 µm often exhibit poor recovery, likely due to incomplete dissection or laser damage, and regions as small as 0.002 mm² exhibit a capture rate of only ~25% (Bury et al, 2022; Chen et al, 2023).

Tissue expansion offers a way to overcome LCM limitations by enhancing resolution and improving sample collection efficiency (Dong et al, 2024; Drelich et al, 2021; Li et al, 2022) (Fig. 1d; Table 1). Filter-aided expansion proteomics (FAXP) exemplifies this strategy, combining optimized expansion with filter-aided in-gel digestion and automation to enhance throughput, reproducibility, and compatibility with formalin-fixed paraffin-embedded (FFPE) tissues. Notably, FAXP is the first method to integrate expansion with LCM for proteomic analysis at the level of individual nuclei (Dong et al, 2024). Beyond MS-based workflows, tissue expansion also supports multiplexed imaging (Kang et al, 2024; Ku et al, 2016) and MSI-based proteomics (Bai et al, 2023). Imaging methods rely on antibody-based detection, either preserving endogenous proteins for conventional immunostaining or using DNA-barcoded antibodies with signal amplification to visualize dozens of targets at nanometer resolution. MSI-based approaches like Expand and comPRESS hydrOgels (ExPRESSO) integrate expansion with high-plex platforms such as imaging mass cytometry (IMC) or multiplexed ion beam imaging (MIBI), achieving subcellular

**Table 1. Expansion omics strategies.**

| Omics category | Strategy | Targeted biomolecule | Anchor | Target specificity | Sample type | Homogenization | Linear expansion factor | Detection method |
|---|---|---|---|---|---|---|---|---|
| Expansion epigenomics | SCEPTRE (Woodworth et al, 2021) | Histone marks, DNA | MA-NHS | Targeted | Cell | Proteinase K, 37 °C, overnight | ~4× | Combine DNA FISH with immunofluorescence and quantify histone mark fluorescence signals |
| | ExEpi (Acke et al, 2022) | Epigenetic readers, histone marks | AcX | Targeted | Cell | Proteinase K, RT, overnight | ~4× | Fluorescent staining |
| | ChromExM (Pownall et al, 2023) | Chromatin | Acrylamide and PFA | Targeted | Zebrafish embryo | SDS, 76 °C, 1–2 h | ~15× | Metabolic labeling of DNA and nascent RNA, along with antibody labeling to visualize the chromatin |
| Expansion transcriptomics | ExFISH (Chen et al, 2016) | RNA | LabelX | Targeted | Cell and mouse brain slices | Proteinase K, 37 °C, overnight | ~3× | smFISH imaging and HCR-amplified FISH imaging |
| | Expansion MERFISH (Wang et al, 2018) | RNA | Acrydite-modified poly(dT) locked nucleic acid probe | Targeted | Cell | SDS, then Proteinase K, 37 °C, >12 h | 2.3× | MERFISH imaging |
| | ExSeq (Alon et al, 2021) | RNA | LabelX | Targeted and untargeted | Cell, neurons, mouse brain, C. elegans, and Drosophila embryos | Proteinase K, 37 °C, overnight | ~4× | Fluorescent in situ sequencing enables untargeted in situ sequencing, Padlock probes allow targeted sequencing, and both use NGS chemistries on a fluorescence microscope. |
| | Ex-ST (Fan et al, 2023) | RNA | Poly(dT) probe | Untargeted | Mouse brain slice | Proteinase K, 37 °C, overnight | ~2.5× | Visium platform |
| Expansion proteomics | proExM (Drelich et al, 2021) | Proteins | AcX | Untargeted | FFPE and FF mouse brain sections | SDS, 58 °C, overnight | ~3× | LC-MS/MS |
| | ProteomEx (Li et al, 2022); FAXP (Dong et al, 2024) | Proteins | NSA | Untargeted | Mouse tissues (Dong et al, 2024; Li et al, 2022), FFPE human CRC samples (Dong et al, 2024) | SDS, 95 °C, 3 h (Li et al, 2022) or SDS, autoclave at 105–121 °C, 60–90 min (Dong et al, 2024) | 5.5–8× (Li et al, 2022); 4–5× (Dong et al, 2024) | LC-MS/MS |
| | ExPRESSO (Bai et al, 2023) | Proteins | Acrylamide | Targeted | Human FFPE tissues (tonsil, tumor, and brain) | SDS, 70 °C for 18 h, followed by 95 °C for 1 h | ~3.7× | MIBI and IMC |
| Expansion lipidomics | GAMSI (Chan et al, 2024), ExIMS (Samuel et al, 2025), Ex-MSI (Hung et al, 2024) | Lipids | AcX | Untargeted | Mouse brain slice | Trypsin, 37 °C, 2–4 days (Chan et al, 2024); Proteinase K, RT, overnight (Samuel et al, 2025); Proteinase K, 60 °C, 3 h (Hung et al, 2024) | ~3–4× (Chan et al, 2024); ~4.5× (Hung et al, 2024; Samuel et al, 2025) | MSI |
| | LExM (White et al, 2022) | Phospholipids | Direct chemical anchoring of metabolically labeled lipids | Targeted | Cell | SDS at 95 °C for 30 min, followed by 80 °C for 2–8 h | 4.9–7.8× | Fluorescence imaging of covalently anchored metabolically labeled lipids |
| | umExM (Shin et al, 2025) | Membrane | pGk13a and AcX | Targeted | Mouse brain slice | Proteinase K, RT, overnight; Trypsin + LysC for staining | ~4× | Fluorescence imaging of covalently anchored, metabolically labeled membranes |

**Table 1.** (continued)

| Omics category | Strategy | Targeted biomolecule | Anchor | Target specificity | Sample type | Homogenization | Linear expansion factor | Detection method |
|---|---|---|---|---|---|---|---|---|
| Expansion multi-omics | uniExM (Cui et al, 2023) | Proteins and RNA | GMA | Targeted | Cell, mouse brain, and PDX breast cancer | Sample-specific homogenization (Proteinase K or SDS) | 4.2–4.4× | Fluorescence imaging of epoxy acrylate-linked proteins and nucleic acids |
| | Dual-ExM (Cho and Chang, 2022) | Proteins and mRNAs | AcX and LabelX | Targeted | Mouse brain tissue and cell | Proteinase K | 4.5× | Fluorescence imaging of antibody-stained proteins and FISH-labeled mRNAs |
| | FLARE (Mao et al, 2020) | Carbohydrates and proteins | Hydrazide- and NHS-functionalized fluorophores | Untargeted | Mouse kidney and FFPE human kidney | 90 °C for 1 h (cell); 70 °C for 2 h, followed by 90 °C for 24 h (tissue) | 5× | Imaging of hydrazide-functionalized fluorophores binding carbohydrates and NHS-functionalized fluorophores labeling proteins |
| | Seq-Scope-X (Anacleto et al, 2025) | Proteins and mRNAs | Poly(dT) probe and DNA-barcoded antibody | Targeted and untargeted | Human and mouse FF organ tissues | SDS + Proteinase K, 37 °C, 24 h | 6–10× | Seq-Scope platform |
| | Click-ExM (Sun et al, 2021) | Nucleic acids, proteins, glycans, lipids, and small molecules | GA and AcX | Targeted | Mouse brain | Proteinase K, 37 °C, 4 h (AcX); 2 h (GA) | ~4.5× | Fluorescence imaging of biomolecules labeled via click chemistry |
| | Magnify (Klimas et al, 2023) | Nucleic acids, proteins, and lipids | Methacrolein | Untargeted | Mouse brain and FFPE human pathology specimens | SDS, 80 °C, 24–72 h; Proteinase K, RT, 2–3 h | ~11× | FISH and fluorescent staining |
| | TEMI (Zhang et al, 2025) | Lipids, metabolites, proteins, and N-glycans | AcX | Untargeted | Mouse cerebellum, melanoma tumor, kidney, and pancreas | Not applied | 2.5–3.5× | MSI |

resolution while enabling broad spatial coverage within tissue sections. Despite these advances, MS-based methods are still limited to selected regions, while imaging and MSI are constrained by antibody specificity, multiplexing limits, and detection sensitivity.

Looking ahead, expansion proteomics strategies like FAXP complement DVP by offering nanoscale visualization and subcellular profiling, thereby uncovering fine spatial organization critical to decoding tissue physiology (Dong et al, 2024; Zheng et al, 2025). In parallel, these strategies enhance AI-based spatial proteomics frameworks such as parallel-flow projection and transfer learning across omics (PLATO) (Hu et al, 2025) and sparse sampling strategy for spatial proteomics (S4P) (Qin et al, 2025) by supplying the physical resolution needed for single-cell analysis across whole slides. By bridging high-resolution sampling with data-driven inference, expansion proteomics contributes to the advancement of scalable, whole-slide, multimodal, single-cell spatial proteomics.

## Expansion lipidomics

Expansion lipidomics can be categorized into non-anchoring and anchoring strategies (Fig. 1e; Table 1). In non-anchoring approaches, MSI-based methods rely on hydrogel expansion to improve spatial resolution while retaining lipids noncovalently through interactions with hydrogel-anchored proteins rather than direct chemical anchoring. Techniques such as gel-assisted mass spectrometry imaging (GAMSI) (Chan et al, 2024), expansion imaging mass spectrometry (ExIMS) (Samuel et al, 2025), and expansion mass spectrometry imaging (Ex-MSI) (Hung et al, 2024) utilize hydrogel-based tissue expansion to achieve cellular or even subcellular resolution in lipid imaging, while maintaining lipid organization. Specifically, GAMSI enhances the MALDI-MSI resolution of both lipids and proteins to sub-micrometer levels, ExIMS improves spatial resolution from ~10–15 μm in unexpanded tissue to ~2.5–3.3 μm in expanded tissue, and Ex-MSI enables the delineation of compact structures with a resolution of about 1 μm. By embedding analytes within the hydrogel, these approaches enhance MSI spatial resolution, minimize lipid delocalization, and maintain molecular

composition, enabling high-resolution lipid visualization without direct lipid anchoring.

In contrast, imaging-based lipidomics employs chemical anchoring to adapt tissue expansion for nanoscale lipid imaging. Ultrastructural membrane expansion microscopy (umExM) enables nanoscale membrane imaging with ~60 nm resolution on a confocal microscope, while maintaining high lipid labeling density and preserving membrane structures in tissues (Shin et al, 2025). Click-expansion microscopy (Click-ExM) utilizes click chemistry for lipid labeling but requires detergent-based permeabilization, which may compromise membrane integrity (Devaraj and Finn, 2021; Sun et al, 2021). Unlike Click-ExM, lipid expansion microscopy (LExM) eliminates the need for detergent treatment and employs a trifunctional chemical linker that covalently attaches lipid molecules to the hydrogel, preserving membrane integrity and enabling super-resolution imaging of nanoscale features beyond the diffraction limit, achieving a resolution of ~100 nm (White-Mathieu and Baskin, 2024; White et al, 2022). This method offers targeted visualization of specific lipids through fluorescence microscopy and facilitates the visualization of nanoscale membrane features.

Together, MSI-based approaches provide non-anchoring lipidomics via protein-mediated retention, typically improving resolution from tens of micrometers down to the micrometer or sub-micrometer scale. Meanwhile, imaging-based methods enable direct lipid anchoring for nanoscale lipid imaging, offering complementary strategies for studying lipid organization in biological systems. However, challenges remain: noncovalent retention in MSI methods may lead to partial lipid loss or redistribution during processing, while anchoring strategies such as LExM may be limited by the availability of broadly reactive linkers for diverse lipid species.

## Expansion multi-omics

Initially applied to single-omics, tissue expansion also supports spatial detection of multiple biomolecules (Dolgin, 2025), enabled by advances in anchoring and labeling strategies (Table 1). For dual-molecule detection, several ExO approaches have been developed. United ExM (uni-ExM) utilizes a single multifunctional anchor (an acrylate epoxide) to covalently

link proteins and RNA, enabling nanoscale visualization with FISH probes and antibodies (Cui et al, 2023). Dual-expansion microscopy (Dual-ExM) combines mRNA and protein labeling via the sequential application of FISH probes and antibodies, followed by covalent anchoring with LabelX and AcX, enabling three-dimensional imaging with nanoscale resolution (Cho & Chang, 2022). Fluorescent labeling of abundant reactive entities (FLARE) further extends this concept by incorporating carbohydrates and proteins through hydrazide- and NHS-functionalized fluorophores that are covalently anchored in the hydrogel, achieving ~65 nm resolution using standard confocal microscopy (Mao et al, 2020). More recently, seq-scope-expanded (Seq-Scope-X) has combined tissue expansion with the Seq-Scope platform (Cho et al, 2021), integrating untargeted transcriptome-wide RNA capture through chip-bound poly(dT) primers with targeted protein detection via DNA-barcoded antibodies (Anacleto et al, 2025). This hybrid strategy reduces transcript diffusion and increases spatial feature density, enabling high-resolution spatial transcriptomics and proteomics at up to ~60 nm, and providing new opportunities to resolve molecular organization and cellular heterogeneity at the nanoscale.

ExO strategies have further evolved to support spatial mapping of multiple molecular classes beyond dual targets. Click-ExM employs metabolic labeling and bioorthogonal click chemistry to selectively anchor molecules, particularly small, difficult-to-retain biomolecules such as lipids and glycans, enabling their spatial visualization following tissue expansion (Sun et al, 2021). Magnify, by contrast, introduces a universal anchoring strategy using methacrolein, allowing broad retention of nucleic acids, proteins, and lipids, and supporting multi-target immunostaining (Klimas et al, 2023). With up to 11× physical expansion, Magnify enables ~25 nm effective resolution on conventional confocal microscopes, which improves to ~15 nm when coupled with super-resolution optical fluctuation imaging. This versatility makes it suitable for integrated multi-omic imaging across diverse and complex tissue types. In contrast to imaging-based ExO approaches, tissue-expansion mass-spectrometry imaging (TEMI) is an MS-based method that enables label-free, multi-molecular spatial mapping of lipids,

metabolites, peptides (proteins), and N-glycans. By avoiding tissue homogenization, TEMI retains these biomolecules in their native spatial context, primarily through their interactions with anchored proteins (Zhang et al, 2025). TEMI achieves high spatial resolution (~2.9 μm), advancing spatial multi-omics beyond conventional imaging.

Despite their distinct molecular anchoring strategies, these methods face common limitations. Achieving uniform retention across biomolecule classes remains difficult and may compromise data completeness. Signal dilution during expansion can reduce sensitivity, especially for low-abundance targets. Moreover, most methods are optimized for optical or mass-spectrometry imaging, but do not support seamless integration with deep proteomic workflows such as advanced LC-MS/MS, thus hindering comprehensive multi-omics analyses. Collectively, ExO enables nanoscale, spatially resolved profiling across multiple omics layers, offering an integrated view of molecular organization in native tissue context.

## Pushing the frontiers of expansion omics

### Resolution enhancement through expansion: How far can we stretch it?

Unlike super-resolution microscopy, which requires specialized and expensive instrumentation (Schermelleh et al, 2019), tissue expansion achieves nanoscale resolution using standard fluorescence microscopes, greatly improving accessibility. The original ExM protocol (Chen et al, 2015) offered ~4.5× linear expansion, enabling ~70 nm lateral resolution. Since then, some efforts have focused on increasing the expansion factor to push resolution further.

Iterative techniques such as iterative expansion microscopy (iExM) (Chang et al, 2017), expansion revealing (ExR) (Sarkar et al, 2022), and pan-ExM (M'Saad and Bewersdorf, 2020) have achieved up to ~13–22× linear expansion, reaching resolutions near 20 nm. Parallel efforts have developed robust single-round protocols such as X10 (Truckenbrodt et al, 2018), ten-fold robust expansion microscopy (TREx) (Damstra et al, 2022), and Magnify (Klimas et al, 2023), attaining 10–11× linear expansion with 25–30 nm resolution. More recently, 20ExM (Wang et al, 2024)

achieved ~20× expansion in a single step, enabling sub-20 nm resolution, while 3D-ExM (Norman et al, 2025) has delivered ~12× isotropic expansion, enhancing resolution in all spatial dimensions. Yet, the current ~20–22× expansion limit should not be viewed as a fundamental barrier. Pushing beyond 30× could yield single-digit nanometer resolution, approaching the size scale of individual protein complexes (Goodsell and Olson, 2000). Achieving this will require novel gel chemistry and embedding strategies that support larger physical magnification while preserving tissue integrity and molecular retention. These innovations would open new possibilities for high-resolution imaging and spatial profiling using standard optical platforms.

However, realizing these gains presents several technical challenges. As expansion factors increase, volumetric growth dilutes biomolecules and reduces detection sensitivity, particularly for low-abundance targets. Achieving isotropic expansion at high magnification can also compromise tissue integrity. Overcoming these limitations will require robust signal amplification, improved molecular stabilization, and precise control of hydrogel uniformity and homogenization. Whether extreme expansion alters the interactions between individual molecules or those in proximity remains to be fully determined.

### Multi-omics integration through selective molecular retention

Tissue expansion has shown potential in enabling high-resolution multi-omics analysis (Table 1). Strategies such as click chemistry (Sun et al, 2021), multi-anchoring (Cho and Chang, 2022), and universal molecular anchoring (Klimas et al, 2023) enable the visualization of molecules such as proteins, nucleic acids, and lipids within the same expanded tissue. While imaging approaches offer high spatial resolution, they remain limited in molecular discovery and detection depth. Profiling-based strategies can complement imaging by expanding molecular coverage while preserving spatial context. Nevertheless, in practice, multi-omics analyses are usually performed sequentially rather than simultaneously (Li et al, 2025), and achieving simultaneous measurement across all major molecular classes, including nucleic acids, proteins, lipids, and small molecules like metabolites, remains rare.

Several challenges exist in implementing profiling-based multi-omics within expanded tissues. First, the sequential separation of distinct molecular types from the hydrogel is technically demanding and risks cross-contamination. Second, certain biomolecules, such as RNA, are prone to degradation during extraction and processing. Third, expansion proteomics remains limited by low detection sensitivity and the absence of high-throughput workflows, making it reliant on targeted rather than unbiased sampling, unlike transcriptomics (Fan et al, 2023) and lipidomics (Chan et al, 2024).

To address these challenges, several strategies remain to be explored. For the separation issue, one approach is the development of reversible anchoring chemistries that allow sequential de-anchoring and collection of different molecular types. Another is the integration of established methods, for example, combining Ex-ST (Fan et al, 2023) for spatial transcriptomics with GAMSI (Chan et al, 2024) for lipidomics, to enable coordinated multi-omics profiling within the same sample. Alternatively, molecular separation could be guided by intrinsic physicochemical properties of biomolecules, bypassing the need for molecular anchoring altogether. To protect labile molecules, particularly RNA, improved stabilization protocols will be essential for maintaining molecular integrity throughout the preparation process. For expansion proteomics, advancing beyond current limitations will require standardized and automated workflows, including voxel-level sampling, robotic integration, and streamlined tissue processing, to improve throughput and reproducibility. Together, these innovations will be critical for building robust and scalable expansion-based multi-omics platforms. Ultimately, bridging imaging- and profiling-based approaches will enable comprehensive, high-resolution spatial profiling across biological systems.

### From suspension to section: adapting expansion omics to diverse sample types

Tissue expansion has been successfully applied to suspension cells, fresh-frozen (FF), and FFPE tissues (Chen et al, 2015; Zhao et al, 2017). Suspension cells are relatively easy to prepare and preserve fine structures well. FF samples maintain native molecular features due to minimal cross-linking, making them highly compatible with multi-molecular detection. FFPE

tissues, widely available in clinical settings and suitable for long-term storage, are valuable resources for clinical and retrospective studies (Bass et al, 2014). Notably, recent studies have demonstrated that FFPE tissue microarrays are compatible with expansion proteomics, enabling simultaneous embedding and standardized processing of multiple clinical samples within a single hydrogel (Hong et al, 2025).

Barriers, however, still exist. Suspension cells must adhere to the culture surface but not too firmly, as excessive adhesion can cause structural disruption. Meanwhile, effective fixation is required to preserve cellular morphology. FFPE tissues are more difficult to expand due to formaldehyde-induced cross-linking and their firm attachment to positively charged glass slides. Additionally, formalin fixation and paraffin embedding can fragment or chemically modify molecules such as nucleic acids, reducing molecular retention and complicating downstream analyses. These issues are often exacerbated in long-term archived FFPE blocks, which tend to show further degradation and diminished sample quality (Bass et al, 2014). Adapting expansion protocols to diverse sample types requires tailored strategies. Approaches include improved adhesion and fixation techniques for embedding cells into hydrogels, optimized collection and storage practices for FFPE sections, and continued advances in hydrogel chemistry and mechanical homogenization to support isotropic expansion. Above all, ensuring high-quality sample preparation from the outset remains fundamental to the success of ExO.

## From brain to bone: expanding into diverse biological systems

Most studies to date have focused on soft tissues such as the brain, which expand uniformly and are relatively straightforward to process. However, there is growing interest in applying expansion approaches to more rigid or heterogeneous specimens, including tumors (Dong et al, 2024), extracellular matrix (ECM)-rich tissues (Chuang et al, 2024; Zhao et al, 2017), calcified structures such as bone (Sim et al, 2025), whole organisms like *C. elegans* (Yu et al, 2020) and planarians (Lim et al, 2019), and even plants (Bos et al, 2024; Hawkins et al, 2023). These efforts aim to capture a broader range of structural contexts and extend applications

beyond neuroscience into fields such as oncology, developmental biology, and plant science. Expanding into these systems presents new technical challenges. Rigid or calcified tissues can impede hydrogel infiltration and resist homogenization, making isotropic expansion more difficult. Diseased or aged tissues introduce additional variability, while whole organs, organisms, and plant structures require longer processing times and more precise control to ensure uniform expansion. Overcoming these barriers will require advances in sample preparation, particularly in molecular anchoring, monomer penetration, and tissue homogenization, to address limited accessibility caused by dense ECM, high calcium content, fibrosis, rigid cell walls, and other structural barriers. Continued innovation in these areas will be essential to unlock the full potential of ExO across structurally diverse tissues and biological systems.

## Computational strategies for high-resolution expansion data

Tissue expansion-based methods such as Ex-ST (Fan et al, 2023), ExPRESSO (Bai et al, 2023), GAMSI (Chan et al, 2024), and TEMI (Zhang et al, 2025) highlight the feasibility of nanoscale, tissue-wide profiling. Tissue expansion combined with STED or SMLM can approach electron microscopy resolution (Fan et al, 2021; Louvel et al, 2023; Shi et al, 2021; Zwettler et al, 2020), while more recent strategies such as one-step nanoscale expansion (ONE) microscopy (Shaib et al, 2024) and integration with deep learning frameworks such as cryoFIRE (Levy et al, 2022) extend this capability toward near-nanometer protein shape reconstruction and tighter linkage to spatial omics maps. However, the increasingly rich datasets generated by these approaches also magnify computational bottlenecks in segmentation, distortion correction, signal deconvolution, and multimodal integration, highlighting the need for dedicated strategies to fully realize the promise of ExO.

Classical segmentation often fails when membranes are discontinuous after tissue expansion, leading to signal misassignment in crowded tissues. To overcome this, a variety of strategies have been explored as potential solutions. High-resolution methods (Heidari et al, 2025; Jones et al, 2025; Petukhov et al, 2021) may improve robustness by integrating molecular signals with morphological priors,

while segmentation-free approaches (Benjamin et al, 2024; Park et al, 2021; Si et al, 2024) offer ways to bypass explicit boundary detection and instead model spatial distributions directly. More recently, graph-based frameworks such as Bering (Jin et al, 2025) have been proposed as a promising direction, leveraging transcript co-localization graphs that could enable more accurate assignment even without nuclear staining.

Second, distortion correction and registration remain critical for reliable nanoscale mapping. Frameworks such as spatial transcriptomics align (STalign) (Clifton et al, 2023), spatial transcriptomics imaging framework (STIM) (Preibisch et al, 2025), and coarSe-to-fine AligNment and sTitching for spatial Omics (SANTO) (Li et al, 2024) illustrate possible avenues for scalable alignment across large fields of view and cross-platform slices, whereas GelMap (Damstra et al, 2023), 3D-aligner (Loi et al, 2024), and multiplexed expansion revealing (multi-ExR) (Kang et al, 2024) introduce intrinsic calibration and nanometer-precision alignment tailored for expansion datasets.

Another persistent challenge is signal deconvolution, which addresses the noise and mixing that arise in densely packed regions. Methods such as SPOTlight (Elosua-Bayes et al, 2021), using seeded non-negative matrix factorization, and Deconvolution of Spatial Transcriptomics profiles using Variational Inference (DestVI) (Lopez et al, 2022), applying variational inference with autoencoder architectures, may help mitigate contamination and provide cleaner estimates of cell- or compartment-specific abundances.

Finally, multimodal integration and prediction present opportunities to connect expansion data with diverse omics layers. De novo spatial reconstruction (NovoSpaRc) (Nitzan et al, 2019) and multi-omics single-cell optimal transport (moscot) (Klein et al, 2025) demonstrate the potential to reconstruct and align spatial maps across time and modalities, while linked inference of genomic experimental relationships (LIGER) (Welch et al, 2019) and spatial integration of multi-omics (SIMO) (Yang et al, 2025) suggest scalable routes for cross-modal integration. In spatial proteomics, frameworks such as PLATO (Hu et al, 2025) and S4P (Qin et al, 2025) apply transfer learning and deep multimodal integration to improve inference; yet their resolution remains above the single-cell level, even with advanced AI strategies. Tissue expansion could provide the nanoscale context needed to accelerate their adaptation to true single-cell resolution.

Moving forward, graph neural networks for spatial modeling, transformer-based multimodal alignment, and generative AI for predictive simulation will be central to advancing ExO from descriptive mapping to mechanistic inference. Together, these AI approaches and broader computational strategies are not simply add-ons; they are foundational. Designing pipelines explicitly optimized for expansion data, benchmarking them systematically, and developing scalable, open-source frameworks will be critical for realizing the full potential of ExO.

## Conclusions and outlook

Expansion microscopy has evolved from a super-resolution imaging method into a versatile platform that bridges nanoscale resolution with multi-omics profiling. By physically decrowding tissues, it enables visualization of chromatin, RNA, proteins, lipids, and other biomolecules in situ, revealing how nanoscale molecular organization underpins processes such as transcriptional regulation, chromatin dynamics, and tissue architecture. These integrated capabilities suggest that ExO could become a cornerstone in the broader landscape of spatial biology.

Future developments should prioritize further increasing linear expansion factors, enhancing molecule-specific anchoring for multiplexed labeling, and optimizing hydrogel systems to accommodate structurally diverse tissues. Parallel advances in automation and high-throughput workflows are expected to be important for improving scalability and accessibility, thereby facilitating the transition of ExO

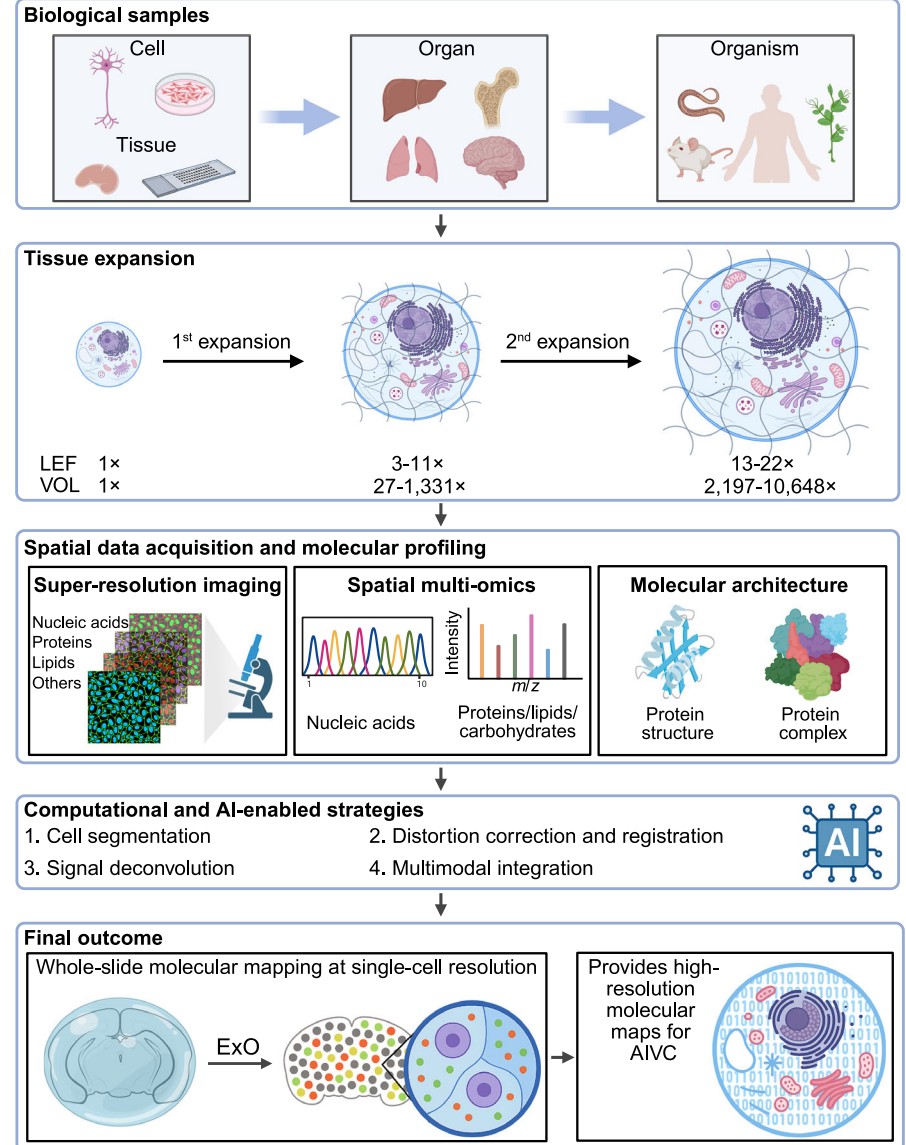

**Figure 2. Expansion omics: from biological samples to AIVC construction.**

Tissue expansion can be applied across diverse biological samples to enhance resolution and enable high-resolution molecular profiling. When combined with computational and AI-enabled strategies, this approach allows whole-slide molecular profiling at single-cell resolution and yields molecular maps for constructing an Artificial Intelligence Virtual Cell (AIVC). LEF linear expansion factor, VOL volumetric expansion factor, ExO expansion omics. Created with BioRender.com.

from specialized laboratories to broader adoption.

As these improvements make expansion datasets more information-rich and multidimensional, they also amplify computational demands. Addressing challenges such as segmentation, distortion correction, registration, signal deconvolution, and multimodal integration will require AI-driven frameworks capable of handling large-scale, heterogeneous datasets. Developing scalable pipelines that adapt existing approaches or introduce new algorithms across these areas will be essential for transforming complex raw inputs into biologically meaningful models.

Looking ahead, the unique capacity of tissue expansion to enable super-resolution imaging and spatial omics within the same sample positions it as a foundational technology for whole-slide, single-cell multi-omics. Together with its ability to reveal detailed molecular architectures, these integrated capabilities could also provide a foundation for future efforts toward constructing the Artificial Intelligence Virtual Cell (Bunne et al, 2024; Qian et al, 2025), which will require spatially resolved imaging, multi-omics data, and nanoscale structural detail to link molecular organization with cellular function in silico (Fig. 2). These innovations have the potential to accelerate biomedical discovery and may help redefine the future of spatial systems biology.

## Declaration of generative AI and AI-assisted technologies in the writing process

During the preparation of this work, the authors used ChatGPT to improve language and readability. After using this tool, the authors reviewed and edited the content as needed and take full responsibility for the content of the publication.

## Peer review information

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

## Acknowledgements

This study is supported by the National Natural Science Foundation of China (Major Research Plan, No. 92259201 to TG), the "Pioneer" and "Leading Goose" R&D Program of Zhejiang (No. 2023C03056 to TG), the State Key Laboratory of Medical Proteomics (No. SKLP-Y202405 to ZD), the National Key R&D Program of China (No. 2021YFA1301600 to TG), and the "Pioneer" and "Leading Goose" R&D Program of Zhejiang (No. 2024SSYS0035 to TG).

## Author contributions

**Zhen Dong**: Conceptualization; Supervision; Funding acquisition; Visualization; Writing—original draft; Writing—review and editing. **Weirong Xiang**: Visualization; Writing—original draft; Writing—review and editing. **Wenhao Jiang**: Visualization; Writing—original draft; Writing—review and editing. **Tiannan Guo**: Conceptualization; Supervision; Funding acquisition; Visualization; Project administration; Writing—review and editing.

## Disclosure and competing interests statement

TG is the shareholder of Westlake Omics (Hangzhou) Biotechnology Co., Ltd. The remaining authors declare no competing interests.

