## [Peer Review File · Molecular Systems Biology]

Expansion Omics: From Expansion Microscopy to Spatial Omics

Zhen Dong, Weirong Xiang, Wenhao Jiang, and Tiannan Guo

Corresponding author(s): Tiannan Guo (guotiannan@westlake.edu.cn), Zhen Dong (dongzhen@westlake.edu.cn)

Review Timeline:	Submission Date:	10th Nov 25
	Editorial Decision:	11th Nov 25
	Revision Received:	12th Nov 25
	Accepted:	13th Nov 25

Editor: Jingyi Hou

Transaction Report:

Please note that the manuscript was previously reviewed at another journal and the reports were taken into account in the decision making process at Molecular Systems Biology. Since the original reviews are not subject to EMBO Press' transparent review process policy, the reports and author response cannot be published.

11th Nov 2025

Manuscript Number: MSB-2025-13459

Title: Expansion Omics: From Expansion Microscopy to Spatial Omics

Author: Tiannan Guo

Zhen Dong

Weirong Xiang

Wenhao Jiang

Dear Tiannan,

Thank you for submitting your Perspective to Molecular Systems Biology. Based on your rebuttal letter, it appears that you have carefully addressed the points raised by the reviewers at the other journal. Overall, we find the manuscript to be very well written, and the topic discussed is interesting, timely, and highly relevant to the field. Therefore, I am pleased to inform you that we will be able to accept it for publication, pending the following editorial-level revisions.

1. Please remove the figures from the manuscript text and upload them as separate high-resolution files. The figure legends should remain in the manuscript.
2. "Summary" should be renamed to "Abstract".
3. "Declaration of interests" should be renamed to "DISCLOSURE AND COMPETING INTERESTS STATEMENT".
4. Would it be possible to reduce the number of references to below 100?
5. Reference format should be formatted according to the Molecular Systems Biology reference style:
 - Citations should be listed in alphabetical order.
 - Please list up to 10 co-authors of a paper before adding et al. in the reference list.
 - Remove DOI for published papers.

Click on the link below to submit your revised paper.

Thank you for submitting this interesting paper to Molecular Systems Biology.

Sincerely,
Jingyi

Jingyi Hou, PhD
Senior Editor
Molecular Systems Biology

*** PLEASE NOTE *** As part of the EMBO Press transparent editorial process initiative (see our Editorial at <https://dx.doi.org/10.1038/msb.2010.72>), Molecular Systems Biology will publish online a Review Process File to accompany accepted manuscripts. When preparing your letter of response, please be aware that in the event of acceptance, your cover letter/point-by-point document will be included as part of this File, which will be available to the scientific community. More information about this initiative is available in our Instructions to Authors. If you have any questions about this initiative, please contact the editorial office (msb@embo.org).

All editorial and formatting issues were resolved by the authors.

MSB-2025-13459R

13th Nov 2025

Dear Tiannan,

Thank you again for sending us your revised manuscript. We are now satisfied with the modifications made and I am pleased to inform you that your paper has been accepted for publication.

Sincerely,
Jingyi

Jingyi Hou, PhD
Senior Editor
Molecular Systems Biology